# Computing and maximizing influence in linear threshold and triggering models

**Justin Khim**
Department of Statistics
The Wharton School
University of Pennsylvania
Philadelphia, PA 19104
jkhim@wharton.upenn.edu

**Varun Jog**
Electrical & Computer Engineering Department
University of Wisconsin - Madison
Madison, WI 53706
vjog@wisc.edu

**Po-Ling Loh**
Electrical & Computer Engineering Department
University of Wisconsin - Madison
Madison, WI 53706
loh@ece.wisc.edu

## Abstract

We establish upper and lower bounds for the influence of a set of nodes in certain types of contagion models. We derive two sets of bounds, the first designed for linear threshold models, and the second more broadly applicable to a general class of triggering models, which subsumes the popular independent cascade models, as well. We quantify the gap between our upper and lower bounds in the case of the linear threshold model and illustrate the gains of our upper bounds for independent cascade models in relation to existing results. Importantly, our lower bounds are monotonic and submodular, implying that a greedy algorithm for influence maximization is guaranteed to produce a maximizer within a $\left(1 - \frac{1}{e}\right)$-factor of the truth. Although the problem of exact influence computation is NP-hard in general, our bounds may be evaluated efficiently. This leads to an attractive, highly scalable algorithm for influence maximization with rigorous theoretical guarantees.

## 1 Introduction

Many datasets in contemporary scientific applications possess some form of network structure [20]. Popular examples include data collected from social media websites such as Facebook and Twitter [1], or electrical recordings gathered from a physical network of firing neurons [22]. In settings involving biological data, a common goal is to construct an abstract network representing interactions between genes, proteins, or other biomolecules [8].

Over the last century, a vast body of work has been developed in the epidemiology literature to model the spread of disease [10]. The most popular models include SI (susceptible, infected), SIS (susceptible, infected, susceptible), and SIR (susceptible, infected, recovered), in which nodes may infect adjacent neighbors according to a certain stochastic process. These models have recently been applied to social network and viral marketing settings by computer scientists [6, 14]. In particular, the notion of influence, which refers to the expected number of infected individuals in a network at the conclusion of an epidemic spread, was studied by Kempe et al. [9]. However, determining an influence-maximizing seed set of a certain cardinality was shown to be NP-hard—in fact, even computing the influence exactly in certain simple models is #P-hard [3, 5]. Recent work in theoretical computer science has therefore focused on maximizing influence up to constant factors [9, 2].

A series of recent papers [12, 21, 13] establish computable upper bounds on the influence when information propagates in a stochastic manner according to an independent cascade model. In such a model, the infection spreads in rounds, and each newly infected node may infect any of its neighbors in the succeeding round. Central to the bounds is a matrix known as the hazard matrix, which encodes the transmission probabilities across edges in the graph. A recent paper by Lee et al. [11] leverages "sensitive" edges in the network to obtain tighter bounds via a conditioning argument. Such bounds could be maximized to obtain a surrogate for the influence-maximizing set in the network; however, the tightness of the proposed bounds is yet unknown. The independent cascade model may be viewed as a special case of a more general triggering model, in which the infection status of each node in the network is determined by a random subset of neighbors [9]. The class of triggering models also includes another popular stochastic infection model known as the linear threshold model, and bounds for the influence function in linear threshold models have been explored in an independent line of work [5, 23, 4].

Naturally, one might wonder whether influence bounds might be derived for stochastic infection models in the broader class of triggering models, unifying and extending the aforementioned results. We answer this question affirmatively by establishing upper and lower bounds for the influence in general triggering models. Our derived bounds are attractive for two reasons: First, we are able to quantify the gap between our upper and lower bounds in the case of linear threshold models, expressed in terms of properties of the graph topology and edge probabilities governing the likelihood of infection. Second, maximizing a lower bound on the influence is guaranteed to yield a lower bound on the true maximum influence in the graph. Furthermore, as shown via the theory of submodular functions, the lower bounds in our paper may be maximized efficiently up to a constant-factor approximation via a greedy algorithm, leading to a highly-scalable algorithm with provable guarantees. To the best of our knowledge, the only previously established bounds for influence maximization are those mentioned above for the special cases of independent cascade and linear threshold models, and no theoretical or computational guarantees were known.

The remainder of our paper is organized as follows: In Section 2, we fix notation to be used in the paper and describe the aforementioned infection models in greater detail. In Section 3, we establish upper and lower bounds for the linear threshold model, which we extend to triggering models in Section 4. Section 5 addresses the question of maximizing the lower bounds established in Sections 3 and 4, and discusses theoretical guarantees achievable using greedy algorithms and convex relaxations of the otherwise intractable influence maximization problem. We report the results of simulations in Section 6, and conclude the paper with a selection of open research questions in Section 7.

## 2  Preliminaries

In this section, we introduce basic notation and define the infection models to be analyzed in our paper. The network of individuals is represented by a directed graph $G = (V, E)$, where $V$ is the set of vertices and $E \subseteq V \times V$ is the set of edges. Furthermore, each directed edge $(i, j)$ possesses a weight $b_{ij}$, whose interpretation changes with the specific model we consider. We denote the weighted adjacency matrix of $G$ by $B = (b_{ij})$. We distinguish nodes as either being *infected* or *uninfected* based on whether or not the information contagion has reached them. Let $\bar{A} := V \setminus A$.

### 2.1  Linear threshold models

We first describe the linear threshold model, introduced by Kempe et al. [9]. In this model, the edge weights $(b_{ij})$ denote the influence that node $i$ has on node $j$. The chance that a node is infected depends on two quantities: the set of infected neighbors at a particular time instant and a random node-specific threshold that remains constant over time. For each $i \in V$, we impose the condition $\sum_j b_{ji} \leq 1$.

The thresholds $\{\theta_i : i \in V\}$ are i.i.d. uniform random variables on $[0, 1]$. Beginning from an initially infected set $A \subseteq V$, the contagion proceeds in discrete time steps, as follows: At every time step, each vertex $i$ computes the total incoming weight from all infected neighbors, i.e., $\sum_{j \text{ is infected}} b_{ji}$. If this quantity exceeds $\theta_i$, vertex $i$ becomes infected. Once a node becomes infected, it remains infected for every succeeding time step. Note that the process necessarily stabilizes after at most $|V|$ time steps. The expected size of the infection when the process stabilizes is known as the influence of $A$ and is denoted by $\mathcal{I}(A)$. We may interpret the threshold $\theta_i$ as the level of immunity of node $i$.

Kempe et al. [9] established the monotonicity and submodularity of the function $\mathcal{I} : 2^V \to \mathbb{R}$. As discussed in Section 5.1, these properties are key to the problem of influence maximization, which concerns maximizing $\mathcal{I}(A)$ for a fixed size of the set $A$. An important step used in describing the submodularity of $\mathcal{I}$ is the "reachability via live-edge paths" interpretation of the linear threshold model. Since this interpretation is also crucial to our analysis, we describe it below.

**Reachability via live-edge paths:** Consider the weighted adjacency matrix $B$ of the graph. We create a subgraph of $G$ by selecting a subset of "live" edges, as follows: Each vertex $i$ designates *at most one* incoming edge as a live edge, with edge $(j, i)$ being selected with probability $b_{ji}$. (No neighboring edge is selected with probability $1 - \sum_j b_{ji}$.) The "reach" of a set $A$ is defined as the set of all vertices $i$ such that a path exists from $A$ to $i$ consisting only of live edges. The distribution of the set of nodes infected in the final state of the threshold model is identical to the distribution of reachable nodes under the live-edges model, when both are seeded with the same set $A$.

## 2.2 Independent cascade models

Kempe et al. [9] also analyzed the problem of influence maximization in independent cascade models, a class of models motivated by interacting particle systems in probability theory [7, 15]. Similar to the linear threshold model, the independent cascade models begins with a set $A$ of initially infected nodes in a directed graph $G = (V, E)$, and the infection spreads in discrete time steps. If a vertex $i$ becomes infected at time $t$, it attempts to infect each uninfected neighbor $j$ via the edge from $i$ to $j$ at time $t + 1$. The entries $(b_{ij})$ capture the probability that $i$ succeeds in infecting $j$. This process continues until no more infections occur, which again happens after at most $|V|$ time steps.

The influence function for this model was also shown to be monotonic and submodular, where the main step again relied on a "reachability via live-edge paths" model. In this case, the interpretation is straightforward: Given a graph $G$, every edge $(i, j) \in E$ is independently designated as a live edge with probability $b_{ij}$. It is then easy to see that the reach of $A$ again has the same distribution as the set of infected nodes in the final state of the independent cascade model.

## 2.3 Triggering models

To unify the above models, Kempe et al. [9] introduced the "triggering model," which evolves as follows: Each vertex $i$ chooses a random subset of its neighbors as *triggers*, where the choice of triggers for a given node is independent of the choice for all other nodes. If a node $i$ is uninfected at time $t$ but a vertex in its trigger set becomes infected, vertex $i$ becomes infected at time $t + 1$. Note that the triggering model may be interpreted as a "reachability via live-edge paths" model if edge $(j, i)$ is designated as live when $i$ chooses $j$ to be in its trigger set. The entry $b_{ij}$ represents the probability that edge $(i, j)$ is live. Clearly, the linear threshold and independent cascade models are special cases of the triggering model when the distributions of the trigger sets are chosen appropriately.

## 2.4 Notation

Finally, we introduce some notational conventions. For a matrix $M \in \mathbb{R}^{n \times n}$, we write $\rho(M)$ to denote the spectral radius of $M$. We write $\|M\|_{\infty,\infty}$ to denote the $\ell_\infty$-operator norm of $M$. The matrix $\text{Diag}(M)$ denotes the matrix with diagonal entries equal to the diagonal entries of $M$ and all other entries equal to 0. We write $\mathbf{1}_S$ to denote the all-ones vector supported on a set $S$.

For a given vertex subset $A \subseteq V$ in the graph with weighted adjacency matrix $B$, define the vector $b_{\bar{A}} \in \mathbb{R}^{|\bar{A}|}$ indexed by $i \in \bar{A}$, such that $b_{\bar{A}}(i) = \sum_{j \in A} b_{ji}$. Thus, $b_{\bar{A}}(i)$ records the total incoming weight from $A$ into $i$. A *walk* in the graph $G$ is a sequence of vertices $\{v_1, v_2, \ldots, v_r\}$ such that $(v_i, v_{i+1}) \in E$, for $1 \leq i \leq r - 1$. A *path* is a walk with no repeated vertices. We define the *weight* of a walk to be $\omega(w) := \prod_{e \in w} b_e$, where the product is over all edges $e \in E$ included in $w$. (The weight of a walk of length 0 is defined to be 1.) For a set of walks $W = \{w_1, w_2, \ldots, w_r\}$, we denote the sum of the weights of all walks in $W$ by $\omega(W) = \sum_i^r \omega(w_i)$.

## 3 Influence bounds for linear threshold models

We now derive upper and lower bounds for the influence of a set $A \subseteq V$ in the linear threshold model.

## 3.1 Upper bound

We begin with upper bounds. We have the following main result, which bounds the influence as a function of appropriate sub-blocks of the weighted adjacency matrix:

**Theorem 1.** *For any set $A \subseteq V$, we have the bound*

$$\mathcal{I}(A) \leq |A| + b_A^T (I - B_{\bar{A}\bar{A}})^{-1} \mathbf{1}_{\bar{A}}. \tag{1}$$

In fact, the proof of Theorem 1 shows that the bound (1) may be strengthened to

$$\mathcal{I}(A) \leq |A| + b_A^T \left( \sum_{i=1}^{n-|A|} B_{\bar{A},\bar{A}}^{i-1} \right) \mathbf{1}_{\bar{A}}, \tag{2}$$

since the upper bound is contained by considering paths from vertices in $A$ to vertices in $\bar{A}$ and summing over paths of various lengths (see also Theorem 4 below). The bound (2) is exact when the underlying graph $G$ is a directed acyclic graph (DAG). However, the bound (1) may be preferable in some cases from the point of view of computation or interpretation.

## 3.2 Lower bounds

We also establish lower bounds on the influence. The following theorem provides a family of lower bounds, indexed by $m \geq 1$:

**Theorem 2.** *For any $m \geq 1$, we have the following natural lower bound on the influence of $A$:*

$$\mathcal{I}(A) \geq \sum_{k=0}^{m} \omega(P_A^k), \tag{3}$$

*where $P_A^k$ are all paths from $A$ to $\bar{A}$ of length $k$, such that only the starting vertex lies in $A$. We note some special cases when the bounds may be written explicitly:*

$$m = 1: \quad \mathcal{I}(A) \geq |A| + b_A^T \mathbf{1}_{\bar{A}} := LB_1(A) \tag{4}$$

$$m = 2: \quad \mathcal{I}(A) \geq |A| + b_A^T (I + B_{\bar{A},\bar{A}}) \mathbf{1}_{\bar{A}} := LB_2(A) \tag{5}$$

$$m = 3: \quad \mathcal{I}(A) \geq |A| + b_A^T (I + B_{\bar{A},\bar{A}} + B_{\bar{A},\bar{A}}^2 - \text{Diag}(B_{\bar{A},\bar{A}}^2)) \mathbf{1}_{\bar{A}}. \tag{6}$$

**Remark:** As noted in Chen et al. [5], computing exact influence is #-P hard precisely because it is difficult to write down an expression for $\omega(P_A^k)$ for arbitrary values of $k$. When $m > 3$, we may use the techniques in Movarraei et al. [18, 16, 17] to obtain explicit lower bounds when $m \leq 7$. Note that as $m$ increases, the sequence of lower bounds approaches the true value of $\mathcal{I}(A)$.

The lower bound (4) has a very simple interpretation. When $|A|$ is fixed, the function $LB_1(A)$ computes the aggregate weight of edges from $A$ to $\bar{A}$. Furthermore, we may show that the function $LB_1$ is monotonic. Hence, maximizing $LB_1$ with respect to $A$ is equivalent to finding a maximum cut in the directed graph. (For more details, see Section 5.) The lower bounds (5) and (6) also take into account the weight of paths of length 2 and 3 from $A$ to $\bar{A}$.

## 3.3 Closeness of bounds

A natural question concerns the proximity of the upper bound (1) to the lower bounds in Theorem 2. The bounds may be far apart in general, as illustrated by the following example:

**Example:** Consider a graph $G$ with vertex set $\{1, 2, \ldots, n\}$, and edge weights given by

$$w_{ij} = \begin{cases} 0.5, & \text{if } i = 1 \text{ and } j = 2, \\ 0.5, & \text{if } i = 2 \text{ and } 3 \leq j \leq n, \\ 0, & \text{otherwise.} \end{cases}$$

Let $A = \{1\}$. We may check that $LB_1(A) = 1.5$. Furthermore, $\mathcal{I}(A) = \frac{n+2}{4}$, and any upper bound necessarily exceeds this quantity. Hence, the gap between the upper and lower bounds may grow linearly in the number of vertices. (Similar examples may be computed for $LB_2$, as well.)

The reason for the linear gap in the above example is that vertex 2 has a very large *outgoing* weight; i.e., it is highly infectious. Our next result shows that if the graph does not contain any highly-infectious vertices, the upper and lower bounds are guaranteed to differ by a constant factor. The result is stated in terms of the maximum row sum $\lambda_{\bar{A},\infty} = \left\| B_{\bar{A},\bar{A}} \right\|_{\infty,\infty}$, which corresponds to the maximum outgoing weight of the nodes in $\bar{A}$.

**Theorem 3.** *Suppose $\lambda_{\bar{A},\infty} < 1$. Then $\frac{UB}{LB_1} \leq \frac{1}{1-\lambda_{\bar{A},\infty}}$ and $\frac{UB}{LB_2} \leq \frac{1}{1-\lambda_{\bar{A},\infty}^2}$.*

Since the column sums of $B$ are bounded above by 1 in a linear threshold model, we have the following corollary:

**Corollary 1.** *Suppose $B$ is symmetric and $A \subsetneq V$. Then $\frac{UB}{LB_1} \leq \frac{1}{1-\lambda_{\bar{A},\infty}}$ and $\frac{UB}{LB_2} \leq \frac{1}{1-\lambda_{\bar{A},\infty}^2}$.*

Note that if $\lambda_\infty = \|B\|_{\infty,\infty}$, we certainly have $\lambda_{\bar{A},\infty} \leq \lambda_\infty$ for any choice of $A \subseteq V$. Hence, Theorem 3 and Corollary 1 hold *a fortiori* with $\lambda_{\bar{A},\infty}$ replaced by $\lambda_\infty$.

# 4 Influence bounds for triggering models

We now generalize our discussion to the broader class of triggering models. Recall that in this model, $b_{ij}$ records the probability that $(i,j)$ is a live edge.

## 4.1 Upper bound

We begin by deriving an upper bound, which shows that inequality (2) holds for any triggering model:

**Theorem 4.** *In a general triggering model, the influence of $A \subseteq V$ satisfies inequality (2).*

The approach we use for general triggering models relies on slightly more sophisticated observations than the proof for linear threshold models. Furthermore, the finite sum in inequality (2) may not in general be replaced by an infinite sum, as in the statement of Theorem 1 for the case of linear threshold models. This is because if $\rho\left(B_{\bar{A},\bar{A}}\right) > 1$, the infinite series will not converge.

## 4.2 Lower bound

We also have a general lower bound:

**Theorem 5.** *Let $A \subseteq V$. The influence of $A$ satisfies the inequality*

$$\mathcal{I}(A) \geq \sum_{i \in V} \sup_{p \in P_{A \to i}} \omega(p) := LB_{\text{trig}}(A), \tag{7}$$

*where $P_{A \to i}$ is the set of all paths from $A$ to $i$ such that only the starting vertex lies in $A$.*

The proof of Theorem 5 shows that the bound (7) is sharp when at most one path exists from $A$ to each vertex $i$. In the case of linear threshold models, the bound (7) is not directly comparable to the bounds stated in Theorem 2, since it involves maximal-weight paths rather than paths of certain lengths. Hence, situations exist in which one bound is tighter than the other, and vice versa (e.g., see the Example in Section 3.3).

## 4.3 Independent cascade models

We now apply the general bounds obtained for triggering models to the case of independent cascade models. Theorem 4 implies the following "worst-case" upper bounds on influence, which only depends on $|A|$:

**Theorem 6.** *The influence of $A \subseteq V$ in an independent cascade model satisfies*

$$\mathcal{I}(A) \leq |A| + \lambda_\infty |A| \cdot \frac{1 - \lambda_\infty^{n-|A|}}{1 - \lambda_\infty}. \tag{8}$$

*In particular, if $\lambda_\infty < 1$, we have*

$$\mathcal{I}(A) \leq \frac{|A|}{1 - \lambda_\infty}. \tag{9}$$

Note that when $\lambda_\infty > 1$, the bound (8) exceeds $n$ for all large enough $n$, so the bound is trivial.

It is instructive to compare Theorem 6 with the results of Lemonnier et al. [13]. The *hazard matrix* of an independent cascade model with weighted adjacency matrix $(b_{ij})$ is defined by
$$\mathcal{H}_{ij} = -\log(1 - b_{ij}), \qquad \forall(i, j).$$
The following result is stated in terms of the spectral radius $\rho = \rho\left(\frac{\mathcal{H} + \mathcal{H}^T}{2}\right)$:

**Proposition 1** (Corollary 1 in Lemonnier et al. [13]). *Let $A \subsetneq V$, and suppose $\rho < 1 - \delta$, where* $\delta = \left(\frac{|A|}{4(n-|A|)}\right)^{1/3}$. *Then $\mathcal{I}(A) \leq |A| + \sqrt{\frac{\rho}{1-\rho}}\sqrt{|A|(n - |A|)}$.*

As illustrated in the following example, the bound in Theorem 6 may be significantly tighter than the bound provided in Proposition 1:

**Example:** Consider a directed Erdös-Rényi graph on $n$ vertices, where each edge $(i, j)$ is independently present with probability $\frac{c}{n}$. Suppose $c < 1$. For any set $|A|$, the bound (9) gives
$$\mathcal{I}(A) \leq \frac{|A|}{1 - c}. \tag{10}$$
It is easy to check that $\rho\left(\frac{\mathcal{H} + \mathcal{H}^T}{2}\right) = -(n - 1)\log\left(1 - \frac{c}{n}\right)$. For large values of $n$, we have $\rho(\mathcal{H}) \to c < 1$, so Proposition 1 implies the (approximate) bound $\mathcal{I}(A) \leq |A| + \sqrt{\frac{c}{1-c}}\sqrt{|A|(n - |A|)}$. In particular, this bound increases with $n$, unlike our bound (10). Although the example is specific to Erdös-Rényi graphs, we conjecture that whenever $\|B\|_{\infty,\infty} < 1$, the bound in Theorem 6 is tighter than the bound in Proposition 1.

# 5   Maximizing influence

We now turn to the question of choosing a set $A \subseteq V$ of cardinality at most $k$ that maximizes $\mathcal{I}(A)$.

## 5.1   Submodular maximization

We begin by reviewing the notion of submodularity, which will be crucial in our discussion of influence maximization algorithms. We have the following definition:

**Definition 1** (Submodularity). *A set function $f : 2^V \to \mathbb{R}$ is* submodular *if either of the following equivalent conditions holds:*

*(i) For any two sets $S, T \subseteq V$,*
$$f(S \cup T) + f(S \cap T) \leq f(S) + f(T). \tag{11}$$

*(ii) For any two sets $S \subseteq T \subseteq V$ and any $x \notin T$, the following inequality holds:*
$$f(T \cup \{x\}) - f(T) \leq f(S \cup \{x\}) - f(x). \tag{12}$$
*The left and right sides of inequality (12) are the* discrete derivatives *of $f$ evaluated at $T$ and $S$.*

Submodular functions arise in a wide variety of applications. Although submodular functions resemble convex and concave functions, optimization may be quite challenging; in fact, many submodular function maximization problems are NP-hard. However, positive submodular functions may be maximized efficiently if they are also monotonic, where monotonicity is defined as follows:

**Definition 2** (Monotonicity). *A function $f : 2^V \to \mathbb{R}$ is* monotonic *if for any two sets $S \subseteq T \subseteq V$,*
$$f(S) \leq f(T).$$
*Equivalently, a function is monotonic if its discrete derivative is nonnegative at all points.*

We have the following celebrated result, which guarantees that the output of the greedy algorithm provides a $\left(1 - \frac{1}{e}\right)$-approximation to the cardinality-constrained maximization problem:

**Proposition 2** (Theorem 4.2 of Nemhauser and Wolsey [19]). *Let $f : 2^V \to \mathbb{R}_+$ be a monotonic submodular function. For any $k \geq 0$, define $m^*(k) = \max_{|S| \leq k} f(S)$. Suppose we construct a sequence of sets $\{S_0 = \phi, S_1, \ldots, S_k\}$ in a greedy fashion, such that $S_{i+1} = S_i \cup \{x\}$, where $x$ maximizes the discrete derivative of $f$ evaluated at $S_i$. Then $f(S_k) \geq \left(1 - \frac{1}{e}\right) m^*(k)$.*

## 5.2 Greedy algorithms

Kempe et al. [9] leverage Proposition 2 and the submodularity of the influence function to derive guarantees for a greedy algorithm for influence maximization in the linear threshold model. However, due to the intractability of exact influence calculations, each step of the greedy algorithm requires approximating the influence of several augmented sets. This leads to an overall runtime of $\mathcal{O}(nk)$ times the runtime for simulations and introduces an additional source of error.

As the results of this section establish, the lower bounds $\{LB_m\}_{m\geq 1}$ and $LB_{\text{trig}}$ appearing in Theorems 2 and 5 are also conveniently submodular, implying that Proposition 2 also applies when a greedy algorithm is employed. In contrast to the algorithm studied by Kempe et al. [9], however, our proposed greedy algorithms do not involve expensive simulations, since the functions $LB_m$ and $LB_{\text{trig}}$ are relatively straightforward to evaluate. This means the resulting algorithm is extremely fast to compute even on large networks.

**Theorem 7.** *The lower bounds $\{LB_m\}_{m\geq 1}$ are monotone and submodular. Thus, for any $k \leq n$, a greedy algorithm that maximizes $LB_m$ at each step yields a $\left(1 - \frac{1}{e}\right)$-approximation to $\max_{A\subseteq V : |A|\leq k} LB_m(A)$.*

**Theorem 8.** *The function $LB_{\text{trig}}$ is monotone and submodular. Thus, for any $k \leq n$, a greedy algorithm that maximizes $LB_{\text{trig}}$ at each step yields a $\left(1 - \frac{1}{e}\right)$-approximation to $\max_{A\subseteq V : |A|\leq k} LB_{\text{trig}}(A)$.*

Note that maximizing $LB_m(A)$ or $LB_{\text{trig}}(A)$ necessarily provides a lower bound on $\max_{A\subseteq V} \mathcal{I}(A)$.

## 6 Simulations

In this section, we report the results of various simulations. In the first set of simulations, we generated an Erdös-Renyi graph with 900 vertices and edge probability $\frac{2}{n}$; a preferential attachment graph with 900 vertices, 10 initial vertices, and 3 edges for each added vertex; and a $30 \times 30$ grid. We generated 33 instances of edge probabilities for each graph, as follows: For each instance and each vertex $i$, we chose $\gamma(i)$ uniformly in $[\gamma_{\min}, 0.8]$, where $\gamma_{\min}$ ranged from 0.0075 to 0.75 in increments of 0.0075. The probability that the incoming edge was chosen was $\frac{1-\gamma}{d(i)}$, where $d(i)$ is the degree of $i$. An initial infection set $A$ of size 10 was chosen at random, and 50 simulations of the infection process were run to estimate the true influence. The upper and lower bounds and value of $\mathcal{I}(A)$ computed via simulations are shown in Figure 1. Note that the gap between the upper and lower bounds indeed controlled for smaller values of $\lambda_{\bar{A},\infty}$, agreeing with the predictions of Theorem 3.

For the second set of simulations, we generated 10 of each of the following graphs: an Erdös-Renyi graph with 100 vertices and edge probability $\frac{2}{n}$; a preferential attachment graph with 100 vertices, 10 initial vertices, and 3 additional edges for each added vertex; and a grid graph with 100 vertices. For each of the 10 realizations, we also picked a value of $\gamma(i)$ for each vertex $i$ uniformly in $[0.075, 0.8]$. The corresponding edge probabilities were assigned as before. We then selected sets $A$ of size 10 using greedy algorithms to maximize $LB_1$, $LB_2$, and $UB$, as well as the estimated influence based on 50 simulated infections. Finally, we used 200 simulations to approximate the actual influence of each resulting set. The average influences, along with the average influence of a uniformly random subset of vertices of size 10, are plotted in Figure 2. Note that the greedy algorithms all perform comparably, although the sets selected using $LB_2$ and $UB$ appear slightly better. The fact that the algorithm that uses $UB$ performs well is somewhat unsurprising, since it takes into account the influence from all paths. However, note that maximizing $UB$ does not lead to the theoretical guarantees we have derived for $LB_1$ and $LB_2$. In Table 1, we report the runtimes scaled by the runtime of the $LB_1$ algorithm. As expected, the $LB_1$ algorithm is fastest, and the other algorithms may be much slower.

## 7 Discussion

We have developed novel upper and lower bounds on the influence function in various contagion models, and studied the problem of influence maximization subject to a cardinality constraint. Note that all of our methods may be extended via the conditional expectation decomposition employed by Lee et al. [11], to obtain sharper influence bounds for certain graph topologies. It would be interesting to derive theoretical guarantees for the quality of improvement in such cases; we leave this

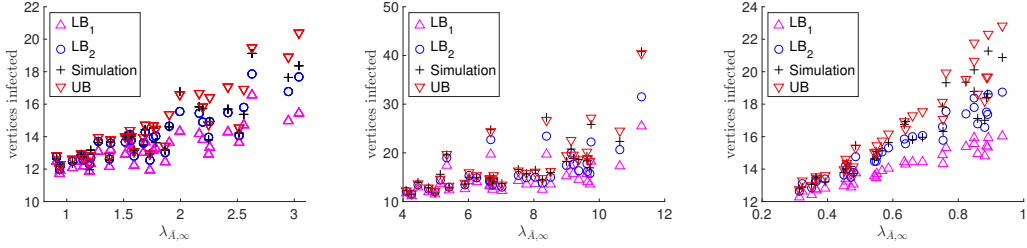

Figure 1: Lower bounds, upper bounds, and simulated influence for Erdös-Renyi, preferential attachment, and $2D$-grid graphs, respectively. For small values of $\lambda_{\bar{A},\infty}$, our bounds are tight.

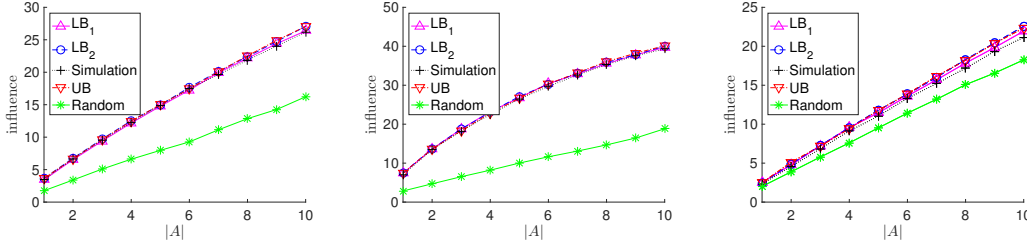

Figure 2: Simulated influence for sets $|A|$ selected by greedy algorithms and uniformly at random on Erdös-Renyi, preferential attachment, and $2D$-grid graphs respectively. All greedy algorithms perform similarly, but the algorithms maximizing the simulated influence and $UB$ are much more computationally intensive.

| | $LB_1$ | $LB_2$ | $UB$ | Simulation |
|---|---|---|---|---|
| Erdös-Renyi | 1.00 | 2.36 | 27.43 | 710.58 |
| Preferential attachment | 1.00 | 2.56 | 28.49 | 759.83 |
| $2D$-grid | 1.00 | 2.43 | 47.08 | 1301.73 |

Table 1: Runtimes for the influence maximization algorithms, scaled by the runtime of the greedy $LB_1$ algorithm. The corresponding lower bounds are much easier to compute, allowing for faster algorithms.

exploration for future work. Other open questions involve quantifying the gap between the upper and lower bounds derived in the case of general triggering models, and obtaining theoretical guarantees for non-greedy algorithms in our lower bound maximization problem.

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
