[Reviews · NeurIPS 2016]

Reviewer 1

Summary

The paper develops new upper and lower bounds for computing the influence of a set in the Linear Threshold cascade model, and in the the General Triggering model, which includes the Independent Cascade model as a special case. These seem to be the first such bounds for the Linear Threshold model, and the bounds for the Independent Cascade model are tighter in some cases than previous results. The analysis is based on fairly elementary "path weight" arguments. These are tighter in the Linear Threshold model, where at most one live path exists from a source to any given node, than in the General Triggering model. The paper analyzes the conditions under which the upper and lower bounds are within a constant factor of each other, and provides examples for which their bounds are tighter than previous ones (ref [16]). The authors further show that the lower bounds are submodular, and so they can be optimized nearly optimally using the greedy algorithm as a principled surrogate for the true influence. Experiments on synthetic graphs demonstrate that the bounds are valid, and that optimizing the bounds as surrogates performs very well for the original influence maximization task.

Qualitative Assessment

The paper is very well written and presents a clear and straightforward analysis of bounds for the influence function in the linear threshold and triggering models. The technical quality is high. The techniques are straightforward (elementary probability arguments) and provide bounds that appear to be useful in practice. The analysis is careful and thorough. For example, Section 3.3 includes both an example of when the bounds are bad, as well as an analysis of conditions under which the bounds are tight. The experiments demonstrate the the bounds are useful in practice for influence maximization. The potential impact is reasonable. There continues to be interest in influence maximization and diffusion processes in the CS literature, and I can envision follow-on work or application of these methods. (Note to authors: in general I feel this line of work has become regrettably detached from real applications, so the extent to which you can connect back to real applications would strengthen future work.) I found two areas that could use further discussion to strengthen the paper: 1) In the experiments on influence maximization, the main benefit of the proposed method is running time. It would be appropriate to compare running times of different methods to quantify the accuracy vs. running-time tradeoff of maximizing the lower bounds. (It appears there is little accuracy lost, so simply reporting running times may tell the story you need to tell). 2) Some further discussion of the comparison to the results of Lemmonier et al. would strengthen the paper. For example, can you quantify empirically the conjecture in Line 217 (condition for when your bound is tighter)? This conjecture suggests that your bound may perform best in the "sub-critical" regime (small cascades relative to graph size). I note that your experiments generally seem to have small cascade sizes (influence < 30 in graphs with 900 nodes). Do your bounds also perform well in with large cascades? Minor notes: Definition 1. The wording is somewhat misleading. It says that "f is submodular if either (i) or (ii)" but does not say that (i) and (ii) are equivalent conditions. Consider adding a statement to this effect for clarity. In Section 5.2 you discuss approximately optimizing the lower bounds, but do not actually state that this problem is NP-hard and cannot be solved optimally. The results in the appendix seem to suggest this. If it is true, I suggest you state it in the main text of the paper.

Confidence in this Review

2-Confident (read it all; understood it all reasonably well)


Reviewer 2

Summary

The paper considers the problems of influence computation and maximization, according to models proposed in the classic work of Kempe, Kleinberg, and Tardos. They consider the threshold model, the independent cascade model, and triggeting models that combine both. The main contribution is deriving lower and upper bounds on the influence I(A) of a set of vertices A from structural properties of the graph and paths in a graph. Since the lower bound function they derive are submodular and monotone, they can be used fto lower bound the maximum influence. Pros: Influence maximization is a very well studied model with applications that go well beyond the original motivations. Presentation is very clear Cons: The results are not very strong. There is a large worst-case gap between the lower and the upper bounds. The motivation and potential applications are not clear. It is fairly simple to obtain some bounds of this form for different influence definitions. These bounds would be much more appealing if the authors could show they more tight for interesting inputs or providing some charactezation that related influence to other notions of centrality of a set of nodes, cuts, or other properties.

Qualitative Assessment

Pros: Influence maximization is a very well studied model with applications that go well beyond the original motivations. Presentation is very clear Cons: The results are not very strong. There is a large worst-case gap between the lower and the upper bounds.

Confidence in this Review

3-Expert (read the paper in detail, know the area, quite certain of my opinion)


Reviewer 3

Summary

The authors present novel upper and lower bounds on influence functions in various contagion models - for the linear threshold model, which are extended to triggering models. They investigate the problem of influence maximization. The lower bounds are monotonic and submodular. Hence, a greedy algorithm for influence maximization is guaranteed to produce a maximizer within a constant approximation factor. The authors report the results of simulations using their methods.

Qualitative Assessment

The paper is well clearly written and well organised. The problem formalisation and the model are interesting and, as far as I know, the results regarding upper and lower bounds on the influence function in various contagion models are novel. I also believe that quantifying the gap between upper and lower bounds in terms of properties of the graph topology represents a significant contribution. Another strength of this paper is represented by the use of a highly scalable algorithm with provable guarantees stemming from the possibility of maximising the provided lower bounds up to a constant-factor approximation. Moreover, these bounds can be extended for the triggering models and, being submodular, it is still possible to provide similar theoretical guarantees for the performance of a greedy algorithm. However, the lower bound expression of Theorem 2 seems to me too implicit. I would have appreciated to see in the general case a connection between this bound and some well known network properties in graph theory. The experimental evaluation is interesting, even if I would have liked to see the results of other simulations running on well known real-world datasets. The results seem to me significant and the used methodology seems rigorous enough. My overall evaluation is positive.

Confidence in this Review

2-Confident (read it all; understood it all reasonably well)


Reviewer 4

Summary

The paper provides an upper bound and a sequence of lower bounds on the final influence of a initial seed set A in the linear threshold model as well as the triggering model. Such bounds are intuitive and novel. For the linear threshold model, the ration between the upper and lower bounds are studied and, when the weights b_{i,j} are symmetric, it is shown that this gap is bounded by a constant (see Corollary 1). Later on, it is shown that one of the lower bounds (LB1) is a monotone submodular set function in terms of the initial set A. Hence, by using a simple greedy algorithm to maximize a submodular function (with cardinality constraints), the authors can find in an efficient manner an initial seed set of cardinality k whose influence is a constant away from the best influence. Numerical simulations on the so called Erdos-Renyi and preferential attachment graphs are given to show the strength os the bounds.

Qualitative Assessment

I believe the results of the paper are novel and elegant. They also lead to an efficient algorithm to maximize influence over initial sets of size k for the linear threshold model. The paper is well written and I enjoyed reading it. I have also gone through the supplementary material to a good depth and I believe the results are correct. Perhaps one issue with the lower bounds is that when the maximum row sum (lambda_{\bar{A}, \infty}) approaches 1, then the difference between the upper and the first order lower bound diverges (see corollary 1). Hence, we need to computed LB_k for large values of k to get a precise lower bound. Also, one main contribution of the paper is that LB_1 is monotone and submodular and hence we can efficiently maximize it given a cardinality constraint. However, if LB_1 is far from the actual value, then this method becomes ineffective. I suspect that this is the case in most networks (i.e. lambda_{\bar{A}, \infty} approaches one for most networks with a large size). Can the authors comment more on this in the paper? (Same issue applies for equations (8) and (9)). Some minor comments: - In the end of Section 2.4, I think the (mathematical) definition of \omega(W) should be corrected. - Between line 373-374, I think the sum should be on the elements of \bar{A} and not the elements of A.

Confidence in this Review

2-Confident (read it all; understood it all reasonably well)


Reviewer 5

Summary

This paper tackles how to bound the influence of a given vertex set under the linear threshold and triggering models, classical standard diffusion models. Specifically, the paper investigates upper and lower bounds in matrix form or spectral properties, and also discusses the gap between these bounds under reasonable assumptions. For the application to influence maximization, the monotonicity and submodularity, which lead to constant approximation via greedy strategy, of the lower bounds for the linear threshold and triggering are established. Experimental evaluations analyze the gap between upper and lower bounds, and the performance of greedy strategy which maximizes the lower bound.

Qualitative Assessment

Technical quality: 3 [proofs] + Proofs for the provided theorems are sound. [results] + Reasonable relaxations which resolve the linearly growing gap are introduced and theoretically discussed. [experiments] + It is verified that the gap between two bounds is not much larger, and greedily maximizing these bounds leads to accurate influence maximization. Novelty / originality: 2 + Upper and lower bounds for the triggering model are general and they can be appropriately applied to other special cases, e.g., the independent cascade model. - I was surprised that the paper did not mention the results of the following articles: [a] Chuan Zhou, Peng Zhang, Jing Guo, and Li Guo. An Upper Bound based Greedy Algorithm for Mining Top-k Influential Nodes in Social Networks. In WWW 2014. [b] Wei Chen, Yifei Yuan, and Li Zhang. Scalable Influence Maximization in Social Networks under the Linear Threshold Model. In ICDM 2010. [c] Zaixin Lu, Lidan Fan, Weili Wu, Bhavani Thuraisingham, and Kai Yang. Efficient influence spread estimation for influence maximization under the linear threshold model. Computational Social Networks 2014. Specifically, the results in the paper were partially established as follows: Eq. (1) in Theorem 1: the same as Eq. (3) in [a]. Eq. (2) in Theorem 1: the same as Eq. (2) in [a]. Eq. (3) in Theorem 2: immediate from the equation in the middle of 4th page of [b]. Also, efficient computation of Eq. (3) in Theorem 2 for small m is discussed in [c]. In summary, the results involving the linear threshold model are lack of novelty. Potential impact or usefulness: 3 + Both influence maximization and influence estimation are well studied, however, most work heavily rely on Monte-Carlo-simulation-based methods. Hence, efficient computation of either upper bound and lower bound is useful. + Suggestion of non-greedy type solution also may lead to the development of new tools in the field of influence propagation. Clarity and presentation: 3 [explanation] + The paper is consistently written, and thus theorems on influence bounds and theoretical properties are easy to follow. + Experimental evaluations briefly describe essentially important observations. [references] - As described above, this paper lacks several articles concerning with upper or lower bounds under the linear threshold models.

Confidence in this Review

2-Confident (read it all; understood it all reasonably well)


Reviewer 6

Summary

In this paper the authors provide bounds (upper and lower) for the influence of set of nodes for some types of contagion models, concretely for the linear threshold model and the triggering model. By using monotonic and submodular properties, they also show that finding the maximizer set of initial infected nodes can be approximated efficiently by greedy algorithms.

Qualitative Assessment

The paper presents new theoretical results on some contagion models. The presentation is very clear and the results are well founded and built on robust mathematical approaches, as the references and previous work on the topic demonstrate. I have no concerns about the paper, but I am not sure about NIPS is the best place to present it. Maybe it will fit better in a more network oriented conference. From the content point of view, I miss some more insights about the practical implications of the results in the paper and maybe som comment about if random paths can be useful to approximate the calculations in a more efficient way.

Confidence in this Review

2-Confident (read it all; understood it all reasonably well)